# A Personalized Collaborative Filtering Recommendation System Based on Bi-Graph Embedding and Causal Reasoning

**DOI:** 10.3390/e26050371

**Published:** 2024-04-28

**Authors:** Xiaoli Huang, Junjie Wang, Junying Cui

**Affiliations:** School of Electrical and Electronic Information, Xihua University, Chengdu 610000, China

**Keywords:** recommendation system, knowledge graph embedding, joint training, collaborative filtering, factorization machine, causal inference

## Abstract

The integration of graph embedding technology and collaborative filtering algorithms has shown promise in enhancing the performance of recommendation systems. However, existing integrated recommendation algorithms often suffer from feature bias and lack effectiveness in personalized user recommendation. For instance, users’ historical interactions with a certain class of items may inaccurately lead to recommendations of all items within that class, resulting in feature bias. Moreover, accommodating changes in user interests over time poses a significant challenge. This study introduces a novel recommendation model, RCKFM, which addresses these shortcomings by leveraging the CoFM model, TransR graph embedding model, backdoor tuning of causal inference, KL divergence, and the factorization machine model. RCKFM focuses on improving graph embedding technology, adjusting feature bias in embedding models, and achieving personalized recommendations. Specifically, it employs the TransR graph embedding model to handle various relationship types effectively, mitigates feature bias using causal inference techniques, and predicts changes in user interests through KL divergence, thereby enhancing the accuracy of personalized recommendations. Experimental evaluations conducted on publicly available datasets, including “MovieLens-1M” and “Douban dataset” from Kaggle, demonstrate the superior performance of the RCKFM model. The results indicate a significant improvement of between 3.17% and 6.81% in key indicators such as precision, recall, normalized discount cumulative gain, and hit rate in the top-10 recommendation tasks. These findings underscore the efficacy and potential impact of the proposed RCKFM model in advancing recommendation systems.

## 1. Introduction

With the continuous development of information technology and the internet we have transitioned from an era of information scarcity to an era of information overload. In addressing this challenge, search engines and recommendation systems have emerged as important technological representatives for mitigating information overload. Traditional search engines primarily satisfy user needs by filtering and screening information, but this approach lacks personalized service. In contrast, recommendation systems can better accommodate user-specific needs, thus becoming a more effective way of information filtering. As a subset of information filtering systems, recommendation systems aim to predict user preferences for products based on their likes, habits, personalized requirements, and product characteristics, thereby recommending the most suitable products to help users make decisions quickly and enhance user satisfaction. However, with the advent of the big data era, traditional recommendation systems face challenges in harnessing the value of data, among which one of the most prominent issues is the cold-start problem caused by data sparsity. To address this challenge, methods based on collaborative filtering are widely considered one of the successful approaches in recommendation systems [1]. Additionally, new attempts have been proposed by some scholars, such as Jiang et al., who constructed a time-aware collaborative filtering model in 2019 [2], and Wulam et al. in 2019 [3] as well as Bai et al. in 2020 [4], who proposed new methods that integrate multiple recommendation algorithms. Some scholars have also considered algorithmic energy efficiency. Effective algorithms have the potential to reduce computational resources and energy consumption, thereby decreasing the energy usage of data centers and devices. For instance, in 2021, Gu et al. [5] explored edge computing, where computational resources are deployed at the network’s edge, allowing efficient processing of computational tasks and data from endpoints through edge nodes. Collaborative edge computing integrates the computational resources of edge nodes to offer more efficient computational services to endpoints, taking into account the computational resource limitations of edge nodes. In 2022, Lin et al. [6] investigated the computational offloading problem in collaborative edge computing networks and optimized computational offloading and resource allocation through collaborative methods. Karabetian et al. [7] also focused on optimizing computational offloading and resource allocation by iteratively analyzing extensive datasets provided by end users using a cloud computing paradigm to enhance the knowledge graph and evaluate energy efficiency and usability.

In the field of recommendation systems, the focus often lies on the connections between users and items, while the consideration of interactions between users and users, as well as items and items, is often overlooked. The emergence of knowledge graphs has provided an effective avenue for the design of recommendation systems in the era of big data. Knowledge graph-based recommendation systems strengthen the semantic information of data by interconnecting users to users, users to items, and items to items, thus enhancing recommendation accuracy. This approach holds significant research significance and practical value, gradually becoming one of the most active branches in the field of recommendation systems. A knowledge graph essentially serves as a specialized knowledge repository (as depicted by the dashed circle in Figure 1), where the entities and relationships contained within it can be embedded to derive features using graph embedding models, thereby enhancing the overall performance of the recommendation system. The integration of recommendation models with knowledge graphs has seen rapid development. For instance, in 2018, Wang et al. [8] introduced a knowledge graph from the news domain into an online news recommendation system, demonstrating the beneficial effects of incorporating knowledge graph embedding as an external knowledge base. The CoFM model proposed by Piao et al. in 2018 [9] also integrates knowledge graph embedding for effective recommendation. The CoFM model combines the TransE [10] graph embedding model with the factorization machine (FM) model, but the TransE model cannot effectively identify 1-N, N-1, and N-N relationships within the knowledge graph. In 2022, Lu et al. [11] proposed the JFMH model, aiming to improve CoFM by integrating TransH graph embedding and the factorization machine (FM) model for collaborative filtering recommendations. However, the TransH model used for graph embedding presents the same issue as the TransE model, wherein entity and relationship vectors are represented in a singular vector space, thus limiting the extraction of entity features. In summary, most existing knowledge graph-based recommender systems suffer from the following problems:Existing graph embedding algorithms struggle with entity- and relation-space representation, resulting in poor graph-embedded features.The current research on knowledge graph-based recommender systems focuses on using knowledge graphs to improve the search capability of recommender systems, using knowledge graphs as a modality for capturing features to provide additional value information [12], and some of the studies consider the complementation of knowledge graphs, with few studies on dealing with the bias problem brought about by graph embedding.Current recommendation algorithms increasingly aim to uncover user interests and provide personalized recommendations. Effectively identifying the volatility of user interests remains a recent challenge in the field of recommendation research.

In this paper, the TransR [13] model was selected for the knowledge graph embedding module, which can better distinguish and express the objective 1-N, N-1, and N-N relationships in the graph network. Additionally, the logic of backdoor adjustment in causal reasoning was employed to repetitively weight the embedded features, eliminating biases introduced by using TransR feature embeddings. For the recommendation module, the factorization machine (FM) model was chosen, which is more effective and accurate in handling extremely sparse feature values compared to traditional collaborative filtering models. Moreover, KL divergence was used in the recommendation module to calculate the diversity of user interests, capturing users with changing interests, facilitating personalized recommendations. Extensive experiments were conducted in this study, and the results demonstrate that the graph representation learning model based on the TransR model can better utilize heterogeneous relationship information in the graph network, that backdoor adjustment logic based on causal reasoning can better eliminate biases introduced during feature learning, and that time-based KL divergence calculation can capture user interaction interests. These methods contribute to improving the performance of top-10 recommendations. This paper integrates the graph embedding TransR model and the factorization machine (FM) [14] model, introducing causal reasoning and KL divergence calculation, to establish a graph representation model based on the TransR model for learning knowledge graphs. Using graph embedding vector features to eliminate biases enhances the recommendation effect of FM. Furthermore, experiments conducted on widely used academic datasets, such as “MovieLens-1M” and “Douban dataset”, demonstrated the effectiveness of our approach.

This paper is structured as follows: Section 1 presents the background of this research, the addressed problem, and the definition of a knowledge graph; Section 2 introduces the research progress in the related field; Section 3 focuses on explaining the RCKFM model proposed in this paper, describing the structure of the model and how each component is implemented; Section 4 describes the dataset used for the experiments in this paper, the chosen baseline model, and the application of evaluation metrics; Section 5 designs replicated experiments to evaluate the superiority of the RCKFM model over multiple baseline models and explores the impact of parameters, such as embedding dimension and training batch size, on recommendation performance; in Section 6, the recommendation model proposed in this paper is summarized and future research objectives are planned.

How to effectively improve the energy consumption of recommender systems is also one of the research issues in this paper. Effective algorithms have the potential to reduce computational resources and energy consumption, thereby decreasing the energy usage of data centers and devices. Optimizing energy efficiency extends device lifespan, enhances operational efficiency, reduces energy costs, and promotes green computing. For instance, in 2021, Gu et al. [5] explored edge computing, where computational resources are deployed at the network’s edge, allowing efficient processing of computational tasks and data from endpoints through edge nodes. Collaborative edge computing integrates the computational resources of edge nodes to offer more efficient computational services to endpoints, taking into account the computational resource limitations of edge nodes. In 2022, Lin et al. [6] investigated the computational offloading problem in collaborative edge computing networks and optimized computational offloading and resource allocation through collaborative methods. Karabetian et al. [7] also focused on optimizing computational offloading and resource allocation by iteratively analyzing extensive datasets provided by end users using a cloud computing paradigm to enhance the knowledge graph and evaluate energy efficiency and usability.

## 2. Related Work

In this section, we will discuss two aspects relevant to our research: collaborative filtering-based recommendation systems and knowledge graph-based recommendation systems.

### 2.1. Collaborative Filtering-Based Recommendation Systems

Currently, one of the most widely used and mature recommendation algorithms is collaborative filtering (CF). The basic idea of collaborative filtering is to recommend items to users based on their past preferences and the choices of other users with similar interests (as shown in Figure 2).

As shown in Figure 2, in collaborative filtering, we use an m×n matrix to express the user’s preference for projects. Generally, the score indicates the user’s preference for the project; the higher the score, the more the user likes the project. A score of 0 indicates that the user has not purchased the project. Each row in the graph represents a user, each column represents a project, and the specific content filled in each cell of the rating table is the user’s rating for the project. The collaborative filtering process involves two steps: one is the prediction process, which predicts the possible rating values of users for non-purchased projects; the other is the recommendation process, which is to recommend the N projects that users are most likely to like based on the results of the prediction stage.

The collaborative filtering recommendation algorithm is mainly based on two conclusions: one is that similar users may have similar preferences for projects; the other is that similarity exists between projects. Specifically, the method based on collaborative filtering mainly mines the similarities between users and users and between projects and projects from the historical interaction data of users and projects, and makes recommendations based on these similarities. The Pearson correlation coefficient formula used during this period is shown in Formula (1):(1)$$\begin{array}{c} Pearson_{x,y}=\frac{cov(x,y)}{\sigma x\sigma y}=\frac{\sum_{i=1}^{n}\left(x_{i}-\bar{x}\right)\left(y_{i}-\bar{y}\right)}{\sqrt{\sum_{i=1}^{n}{\left(x_{i}-\bar{x}\right)}^{2}}\sqrt{\sum_{i=1}^{n}{\left(y_{i}-\bar{y}\right)}^{2}}} \end{array}$$
where cov is the calculated covariance, $cov\left(x,y\right)=\frac{1}{n-1}\sum_{i=1}^{n}\left(x_{i}-\bar{x}\right)\left(y_{i}-\bar{y}\right)$, σx is the calculated standard deviation of *x*, $\sigma_{x}^{2}=\frac{1}{n-1}\sum_{i=1}^{n}{\left(x_{i}-\bar{x}\right)}^{2}$, and $\bar{x}$ is the mean value of *x*, $\bar{x}=\frac{1}{n}\sum_{i=1}^{n}x_{i}$.

In recent years, many improvements have been made to collaborative filtering recommendation algorithms. In 2010, Rendle et al. [14] proposed the factor decomposer FM model, which is based on logistic regression and adds a second-order feature crossover part to the model, which is able to better mine the correlation between data features under highly sparse conditions, especially for the crossover data that did not appear in the training samples. Meanwhile, FM can be used in computing the objective function, and doing optimization learning in stochastic gradient descent can be done in linear time. In 2015, Sedhain et al. [15] proposed to take the rating records of each item from all users as inputs and learn the latent representations of the items in order to reconstruct the predicted preferences of each item from all users. In 2016, Blondel et al. [16] proposed that higher-order FM could be used as a model to reconstruct the predicted preferences of each item from all users by direct extension of second-order interactions with all feature interactions, but the modeling was still affected by noisy feature interactions. In 2018, Piao et al. [9] proposed the CoFM model, fusing the TransE model and the factorizer FM model for recommendation. In 2022, Lu et al. [11] proposed JFMH, fusing graph embedding TransH and collaborative filtering for recommendation.

### 2.2. Recommendation System Based on Knowledge Graph

Knowledge graphs can be seen as heterogeneous networks, including various entities and relationships, which are interconnected to build a network that displays comprehensive knowledge information in a certain field. For example, in Figure 1, the characters and movies circled in dotted lines are different types of entity nodes (such as actors and movies, or types and movies). The dashed lines in the figure without nodes represent various relationships between these entities (for example, “its actors” indicates the relationship between actors and movies, and “its type” indicates the relationship between type and movie). For the relationships in Figure 1, we define them in the form of triplets: (Leonardo DiCaprio, its actor, The Basketball Diaries), (Love, its type, Love at First Sight). The recommendation system can use the external supplement library composed of entities and relationships in the knowledge graph to mine the possible relationships between users and projects [17,18]. Specifically, for User A in Figure 1, we know he likes the movie “The Basketball Diaries” starring Leonardo DiCaprio. Based on the two relationships in Figure 1, namely, (Leonardo DiCaprio, its actor, The Basketball Diaries) and (Leonardo DiCaprio, its actor, Titanic), we can infer that User A might also like the movie “Titanic”. Therefore, we can recommend the movie “Titanic”, which was not originally included in the historical interaction data, to User A. In the same way, we can also recommend “The Basketball Diaries” and “Love at First Sight” to User B, making the recommendation process more flexible.

In the recommendation system based on the knowledge graph, the feature learning of the knowledge graph is mainly to obtain a low-dimensional vector through learning entities and relationships. Feature learning of knowledge graphs includes two types of models: one is a distance-based model and the other is a semantic matching model. The focus of this article is on distance-based models. Recently, knowledge graph-based recommender systems have been developing rapidly. In 2018, Wang et al. [8] proposed the DKN model for online news recommender systems. In 2018, Piao et al. [9] proposed the CoFM model to capture knowledge graph features using TransE-assisted recommender systems. In 2020, Wang et al. [19] proposed the CKAN model by exploring the message-passing mechanism on knowledge graphs that exploit higher-order connectivity in an end-to-end manner. Meanwhile, in 2023, our team [20] proposed a deep hash-embedded recommendation based on a knowledge graph, and the model had very good results.

The study of graph embedding algorithms originated from an in-depth analysis and understanding of word vectors, as shown in Formula (2):(2)$$\begin{array}{c} Vec(King)-Vec(Man)=Vec(Queen)-Vec(Woman) \end{array}$$

Word vectors have the ability to perform space translation. Therefore, graph embedding of entities and relationships in a knowledge graph can effectively provide external knowledge bases. The graph embedding methods based on the Trans series are very important parts in the graph embedding algorithm, including basic models such as the TransE model, TransH model, and TransR model (as shown in Figure 3). This method has two main advantages: one is that it can directly embed entities and relationships into low dimensions, and the other is that it can better discover hidden relationships in the graph.

Figure 3a–c, respectively, correspond to three distance-based translation models. In these models, the head node *h*, relationship *r*, and tail node *t* of the triple all have corresponding vector representations, and our expectation is that *h* + *r* = *t*. The closer the result of the vector *h* + *r* is to *t*, it indicates that these vectors can well represent the entities and relationships in the knowledge graph. It can be seen that the improvement of the TransR graph embedding model compared to TransE is that it projects the relationship *r* into a dedicated relationship vector space, which is explained in detail in Section 3.1. TransE is the basic starting algorithm in the Trans series of methods. Because its entities and relations share the same space, it cannot handle types of problems, such as 1-N, N-1, and N-N. To illustrate this problem, for example, in the knowledge graph of Figure 1, there are two triples: (Leonardo DiCaprio, his profession is an actor, his movie is “The Basketball Diaries”) and (Leonardo DiCaprio, his profession is an actor, his movie is “Titanic”). With the idea of the TransE model, when we try to learn the node embedding of Leonardo DiCaprio, it is found that the optimal vector representation of the head entity Leonardo DiCaprio cannot be obtained; this is because the tail entities “The Basketball Diaries” and “Titanic” belong to two different types of movies (one is crime genre, the other is romance–disaster film). For instance, when using the triple (Leonardo DiCaprio, actor, The Basketball Diaries) for training, we get a vector representation of the node Leonardo DiCaprio. However, when (Leonardo DiCaprio, actor, Titanic) is used for testing, we find that the vector representation of the Leonardo DiCaprio node has changed, so the model needs to be optimized in the direction of (Leonardo DiCaprio, his profession is an actor, his movie is “Titanic”), thus causing conflicts in the vector representation for the node of Leonardo DiCaprio. In this case, we considered using the TransR graph embedding model to improve the complete match matrix (CoFM).

## 3. Personalized Collaborative Filtering Recommender System Based on Bipartite Graph Embedding and Causal Inference

In this paper, we take the CoFM fusion model as a reference, which incorporates the TransE graph embedding model and the factorizer FM model. Because of the TransE embedding model, the model cannot solve the 1-N, N-1, and N-N problems well; at the same time, there is the influence of feature bias in the training time, and the predicted results cannot perform the adaptive recommendation to meet the user’s personalized needs. In order to solve these problems effectively, we designed R the CKFM (recommendation algorithms based on TransR, causal reasoning, KL scatter and factorization machine) model. The core work that needs to be done is as follows:Improve the ability of the CoFM model to recognize 1-N, N-1, and N-N relationships. Specifically, we hope to improve the effectiveness of the CoFM model by solving the problem that TransE cannot recognize 1-N, N-1, and N-N relations well. Here, we use the TransR graph embedding model instead of the TransE model, which is able to jump out of the single space when embedding and to deal with many-to-many relations more effectively and accurately.Reconstructing the user’s eigenvalues using backdoor adjustment of causal inference to eliminate the biased effect of user–item-history interactions on the user’s eigenvalues.Using symmetric KL scatter to fuse the traditional FM recommendation with the final predictive score of RCKFM to achieve an adaptive personalized user–item recommendation.

The flow of the RCKFM model recommendation is shown in Figure 4.


where the blue part represents the feature-learning section, and the green part denotes the recommendation algorithm’s portion. Notably, the functionally concentrated steps in the RCKFM model consist of TransR bipartite graph embedding, backdoor adjustment for bias elimination, FM collaborative filtering, and symmetric KL scatter score aggregation. The overall framework of the RCKFM model is shown in Figure 5:


The model structure is divided into four parts:TransR dual graph embedding: we take the knowledge graph dataset and the user–item-history interaction dataset as the input of the whole system. The graph embedding uses the TransR model based on graph embedding to represent the structured information in the knowledge graph. At the same time, to obtain the movie embedding and user embedding, here we consider dual graph embedding for users and projects. The specific operation steps are to constitute the graph embedding of the attributes and types of the movie at the same time as constituting the graph embedding of the user’s attributes and history interactions, etc., in order to obtain the project feature representations and user characterization of the project features and user features, respectively. The TransR graph embedding model is utilized to jump out of the TransE graph embedding model single-space (entity and relationship shared space) representation problem.Causal inference backdoor adjustment: After obtaining the embedding results, we adjust the representation of user-feature vectors by the causal inference backdoor adjustment method to eliminate the bias influence of user-history interaction on user-features representation.FM collaborative filtering: collaborative filtering will be the user–item-history interaction relationship of data collection after matching the embedding results as the input of the FM model. Here, the collaborative filtering model FM should be trained both before the backdoor adjustment of the parameters and after the backdoor adjustment of the parameters to facilitate the personalized scoring fusion.KL Scatter Score Aggregation: Using symmetric KL scatter, we calculate whether the user’s interest is variable according to the timestamp, we fuse the predicted scores before and after the backdoor recommendation, and, ultimately, we obtain the top-10 recommendation list according to the order of the predicted scores.

### 3.1. TransR Dual Graph Embeddings

To solve the representation problem of TransE and TransH, Yankai proposed the TransR [13] model (as shown in Figure 6), in 2015, for the representation of entities and relations in different semantic spaces. It embeds entities and relations in different spaces and solves the problem that they are difficult to represent in a common semantic space. In contrast, TransE and TransH assume that entities and relations are embedded in the same space, but in reality they are different objects that are not necessarily suitable for representation in the same semantic space. Although TransH improves flexibility by introducing relational hyperplanes, it does not completely break this assumption. To overcome the problem of a single space, TransR provides different entity and relation spaces and learns embeddings by translating in the relation space to better represent associations between entities and relations.

The TransR model uses a different embedding method, but unlike TransE it does not directly superimpose the head entity vectors on the relation vectors. Rather, for each triangular group (*h*, *r*, *t*), TransR projects the head entity *h* in the entity space to the relation space using the matrix $M_{r}$ to obtain $h_{r}$. Similarly, the tail entity *t* is projected to the relation space by the matrix $M_{r}$ to obtain $t_{r}$. What is obtained is the embedding representation in the relation-specific space, as shown in Equation (Equation 3). Based on this, the TransR model is represented as $h_{r}$ + *r* ≈ $t_{r}$ by constraining $h_{r}$, $t_{r}$ with the relation vector *r*. As with TransE, the embedding vectors of entities and relations are learned by minimizing the loss function. Here, *h* and *t* denote the embedding vectors of entities and r denotes the embedding vector of relations:(3)$$\begin{array}{c} h_{r}=hM_{r},t_{r}=tM_{r} \end{array}$$

The projection specific to the relationship can cause the head/tail entities that hold this relationship (represented as colored circles in Figure 6) to be close to each other, while those entities that do not hold this relationship are far from each other (represented as colored triangles in Figure 6). Also, the TransR model defines the loss function *L*, as shown in Equation (Equation 5):(4)$$\begin{array}{c} f_{r}\left(h,t\right)={||h_{r}+r-t_{r}||}_{2}^{2} \end{array}$$
(5)$$\begin{array}{c} L=\sum_{(h,r,t)\in S}\sum_{\left(h^{\prime },r^{\prime },t^{\prime }\right)\in S^{\prime }}max\left(0,f_{r}\left(h,t\right)+\gamma -f_{r}\left(h^{\prime },t^{\prime }\right)\right) \end{array}$$
where $S^{\prime }$ represents the negative sample set, negative samples are constructed by randomly replacing either the head entity or the tail entity in the original triplet, and γ is a constant that represents the maximum distance between positive and negative samples.

In the preliminary of the RCKFM model, we constructed a user-attribute knowledge graph and a movie-attribute knowledge graph based on the dataset. We need to obtain the feature values of the corresponding users and movies in the dimensions, so we consider using the graph-based translation distance model. The historical interactions of users are incorporated when constructing the user knowledge graph to embed users so as to obtain more comprehensive user representation (as shown in Figure 7a), while the movie knowledge graph is reconstructed to obtain accurate movie representation (as shown in Figure 7b), and the user and movie graphs are embedded in a dual graph:

Through the dual graph embedding of the TransR model, we can effectively extract the multi-dimensional features of movies and users. The dimension is set in the {16, 32, 64, 128} range, which facilitates model training and also makes it convenient to calculate the loss value of the embedding.

### 3.2. Backdoor Adjustment of Causal Inference

When entity feature vectors are derived through embedding from entities, attributes, and historical interactions, these features often exhibit bias. To address these biases, we propose the use of causal inference. The aim of causal reasoning is to uncover causal relationships between events, variables, or factors, beyond merely their correlations. Backdoor adjustment, a significant concept in causal inference, can mitigate biases inherent in feature embeddings. Given that feature embeddings might contain potentially confounding variables impacting our understanding of causality, backdoor adjustment methods are routinely used to tackle biases arising from unobserved variables, thereby improving the accuracy of causal effect inference. Backdoor adjustment involves two primary steps: the identification of the backdoor path and the control of confounding variables along this path. Here, backdoor paths refer to all trajectories from the dependent variable towards the effect variable.

In this study, we apply the backdoor adjustment of causal inference to a movie graph-based recommendation system [21], focusing mainly on user features representation (U), item features representation (I), user’s preference features for different item categories (M), users’ historical interactions with items (D), and predicted score (Y). U, I, and M are feature vectors, whereas D can be click-through rates or ratings, etc. In the TransR embedding process, a user’s historical interactions D directly affect user features U and user’s preference features M for different item categories, while user features U directly affect user’s preference features M for different item categories. The ultimate predicted score is directly determined by U, I, and M. According to this logic, there are two backdoor paths U<-D->M and M<-U->Y. In this process, the main adjustment is embedding of user features U, so the mediation of M does not need to be considered, only the path U<-D->M. We could opt to block D->U or D->M, but since M is computed from a mix of user features representation U and user’s historical interactions D, its value is difficult to estimate directly. Thus, the most convenient backdoor path adjustment would be to block D->U, as shown in Figure 8:

For U, denoted as $u=[u_{1},\ldots ,u_{K}]$, it is a K-dimensional user-feature embedding vector with $u_{K}$ as a specific feature embedding value. For I, denoted as *i*, it is similar to *u* and it is also an item-feature embedding vector. For D, when the user is *u*, for the item group $\left\{g_{1},\ldots ,g_{N}\right\}$, $d_{u}=\left[p_{u}\left(g_{1}\right),\ldots ,p_{u}\left(g_{N}\right)\right]$ it is a specific value of D, where $p_{u}(g_{N}$) represents historical interactions with the item. For M, it is a vector that describes the user’s preference levels for different types of items. When *d* and *u* are determined, M is also determined. We express this using the function M(*d*,*u*). Many recommendation models can simulate M, as they have explicitly or implicitly modeled the user’s preferences for item categories.

Due to the presence of confounding factors, the current recommendation models that estimate the conditional probability P(Y|U,I) may face the issue of spurious correlation, which can lead to amplified bias. Therefore, given U = *u* and I = *i*, we can derive the conditional probability P according to the following Equations (6)–(9):(6)$$\begin{array}{cc} P(Y|U=u,I=i)= & \frac{\sum_{d\in D}\sum_{m\in M}P\left(d\right)P\left(u|d\right)P\left(m|d,u\right)P\left(i\right)P\left(Y|u,i,m\right)}{P(u)P(i)} \end{array}$$(7)$$\begin{array}{cc} = & \sum_{d\in D}\sum_{m\in M}P\left(d|u\right)P\left(m|d,u\right)P\left(Y|u,i,m\right) \end{array}$$(8)$$\begin{array}{cc} = & \sum_{d\in D}P\left(d|u\right)P\left(Y|u,i,M\left(d,u\right)\right) \end{array}$$(9)$$\begin{array}{cc} = & P\left(d_{u}|u\right)P\left(Y|u,i,M\left(d_{u},u\right)\right) \end{array}$$
where Equation (Equation 6) is introduced by the causal diagram (a) and Equation (Equation 7) is introduced by Bayes’ theorem. When *d* and *u* are deterministic, *m* = M(*d*,*u*), and M denotes a function, then P(M(*d*,*u*)|*d*,*u*) sums to 1, introducing Equation (Equation 8). When *u* is deterministic, *d* is also deterministic, and $d_{u}$ denotes the history of the interaction data corresponding to *u*, introducing Equation (Equation 9). For the data that are not *u*, P(*d*|*u*) is naturally 0.

According to the backdoor adjustment theory, the target formula is P(Y|do(U = *u*), I = *i*). Here, do(U = *u*) can be understood as cutting off the related edges in the causal graph, thus preventing potential effects, as shown in Figure 8b. Its formula representation is
(10)$$\begin{array}{cc} P(Y|do(U=u),I=i)= & \sum_{d\in D}P\left(d|do\left(U=u\right)\right)P\left(Y|do\left(U=u\right),i,M\left(d,do\left(U=u\right)\right)\right) \end{array}$$
(11)$$\begin{array}{cc} = & \sum_{d\in D}P\left(d\right)P\left(Y|do\left(U=u\right),i,M\left(d,do\left(U=u\right)\right)\right) \end{array}$$
(12)$$\begin{array}{cc} = & \sum_{d\in D}P\left(d\right)P\left(Y|=u,i,M\left(d,u\right)\right) \end{array}$$
where Equation (Equation 10) is the same as Equation (Equation 8), Equation (Equation 11) is introduced by the insertion and deletion of actions in the DO algorithmic rules, and Equation (Equation 12) is introduced by the transformation of actions and observations in the DO algorithmic rules.

With the backdoor adjustment, it can be found that the original $d_{u}$ becomes *d*, which now relies on the prior distribution of d and no longer on the conditional distribution of *u*. Therefore, the recommendation of *i* to user *u* is no longer affected by the higher-rated item i of user u in $d_{u}$. This greatly alleviates the problems caused by data bias. Due to the large range of D, for Equation (Equation 12) only the interacted D is considered, which is expressed formulaically as
(13)$$\begin{array}{cc} P(Y|U=u,I=i)= & \sum_{d\in D}P\left(d\right)P\left(Y|=u,i,M\left(d,u\right)\right) \end{array}$$
(14)$$\begin{array}{cc} \approx & \sum_{d\in D}P\left(d\right)f\left(u,i,M\left(d,u\right)\right) \end{array}$$
(15)$$\begin{array}{cc} = & f(u,i,M\left(\sum_{d\in D}P\left(d\right)d,u\right)) \end{array}$$
(16)$$\begin{array}{cc} = & f(u,i,M\left(\bar{d},u\right)) \end{array}$$
where f(·) represents the common recommendation model. Knowing M is enough to implement backdoor adjustment. If the function M can be implemented, the current recommendation model can use it as an additional input to accomplish backdoor adjustment. We can use FM to solve for M, thereby obtaining Formula (17), which is formally represented as
(17)$$\begin{array}{cc} M(\bar{d},u)= & \sum_{a=1}^{N}\sum_{b=1}^{K}p\left(g_{a}\right)v_{a}⨂x_{u,b}u_{b} \end{array}$$
(18)$$\begin{array}{cc} = & \sum_{a=1}^{N+K}\sum_{b=1}^{N+K}w_{a}c_{a}⨂w_{b}c_{b} \end{array}$$
where the function M represents user representation with respect to categories, indicating user preferences in different categories, ⨂ represents element-wise multiplication, and $w=\left[\bar{d},x_{u}\right]=\left[p\left(g_{1}\right),\ldots ,p\left(g_{N}\right),x_{u,1},\ldots ,x_{u,K}\right],c=\left[v,u\right]=\left[v_{1},\ldots ,v_{N},u_{1},\ldots ,u_{K}\right]$. Here, *u* is the user representation, $x_{u}$ is the feature value, and *v* represents the item category.

Using causal inference to eliminate bias is actually about repeatedly weighting the embedded entity features on the dimension of entity attributes, so as to discover better entity features.

### 3.3. Factorization Machine (FM) and Collaborative Filtering

Collaborative filtering models often take the characteristics of users and projects, as well as the results of interactions, as inputs to the model. Therefore, the RCKFM model considers using the rating triplet of user–project historical interactions to achieve feature-matching input and model training. The specific process is shown in Figure 9:

We use *u* in the historical rating triad to match the *u* feature vector obtained from the TransR embedding, and *i* to match the i feature vector with *i*. A higher-level collaborative filtering model is used to fit the user and the movie and predict the final rating. In this case, a collaborative filtering-based factorizer FM model is used, assuming that the scenario is to predict the user’s rating of the movie (as shown in Figure 10), and the user’s rating of the movie is predicted, with each row representing a sample. The data can be divided into two parts: user features and movie features. And the target value *y* denotes the rating. Usually the rating triad constitutes a very sparse sample for training. The key of the FM model is to solve the problem of learning the combined feature weights under sparse data, and to be able to learn the weight information of a specific combination of features from the user features and the movie features, so as to perform the prediction of movie ratings more accurately. Such combined feature weight learning can cope well with the sparsity of data and improve the performance and accuracy of the recommender system.

Currently, each column represents one feature. Suppose we need to build a model from x to y and need to use second-order polynomial features, i.e., we need to use pair combinations of feature x. Therefore, the second-order definition of FM is shown in Formula (19) [9]:(19)$$\begin{array}{c} {\hat{y}}_{FM}\left(x\right)=w_{0}+\sum_{i=1}^{n}w_{i}x_{i}+\sum_{i=1}^{n-1}\sum_{j=i+1}^{n}\hat{w_{ij}}x_{i}x_{j} \end{array}$$
where ${\hat{y}}_{FM}\left(x\right)$ is the final prediction function of the FM algorithm idea. For each given feature vector denoted by $x,x\in R^{n}$, ${\hat{y}}_{FM}\left(x\right)$ can give a prediction score; $w_{0}\in R$ is a constant term, $w_{i}\in R^{n}$ denotes the weight parameter of the deviation term (i.e., the first-order eigencoefficient), and $\hat{w_{ij}}$ is the second-order eigencoefficient, which is defined as shown in Equation (Equation 20). FM learns a one-dimensional vector of size *k* for each feature, and the weight value of the combination of the features of two features $x_{i}$ and $x_{j}$ can be expressed by the inner product of the vectors $v_{i}$ and $v_{j}$ corresponding to the features. This expression corresponds to a matrix decomposition,
(20)$$\begin{array}{c} \hat{w_{ij}}=v_{i}^{T}v_{j}:=\sum_{l=1}^{k}v_{il}v_{jl} \end{array}$$
where *v* is an n×k vector matrix, *n* is the number of features *x*, and *k* is a parameter to be determined, $v_{i}={\left(v_{i1},v_{i2},\cdots ,v_{ik}\right)}^{T}\in R^{n\times k},i=1,2,\cdots ,n,k\in N^{+}$. From Equation (Equation 20), the weight parameters of the second-order terms in the FM model are obtained by multiplying the two matrices $v_{i}$ and $v_{j}$. Therefore, FM can effectively deal with the highly sparse feature data and still estimate the parameters for components that do not interact in the sample. The loss function of the FM model, for which we refer to the work of Noia et al. [22], is defined as shown in Equation (Equation 21):(21)$$\begin{array}{c} L_{p}=\sum_{a^{+}\in y^{+}}\sum_{a^{-}\in y^{-}}-log\left[\delta \left(a^{+}-a^{-}\right)\right] \end{array}$$
where δ is the sigmoid function. The sample sets are the positive sample set and the constructed negative sample set.

The FM algorithm does not depend on whether a particular combination of features has appeared or not, because it learns the embedding vectors of individual features. As long as feature $x_{i}$ and any other feature combination have appeared, their embedding vectors can be learned. Therefore, even though the feature combination $x_{i}$ and $x_{j}$ has not appeared in the training data, if this new feature combination is encountered during prediction the weight of this new feature combination can be calculated by the inner product because both $x_{i}$ and $x_{j}$ have corresponding embedding vectors. This feature makes the FM model highly generalizable. It is able to predict unseen feature combinations by making inferences based on already-learned feature vectors.

With the FM model, we obtain the user’s predicted ratings for the movie along with the value of the loss resulting from collaborative filtering, and in the co-training of the model we need to fuse this loss with the one obtained from the graph embedding in Section 3.1.

One issue with joint training is how to merge the losses of the model. According to the two-task transfer learning in the CoFM model proposed by Piao et al. in 2018, one problem with transfer learning between the two tasks is the different output scales of the two loss functions (Equations (5) and (21)). Therefore, we modify the loss function of the knowledge graph completion task (Equation (Equation 5)) to be as shown in Equation (Equation 22) in order to have the same scale for the loss functions in both tasks:(22)$$\begin{array}{c} L=\sum_{(h,r,t)\in S}\sum_{\left(h^{\prime },r^{\prime },t^{\prime }\right)\in S^{\prime }}-log\left[\delta \left(max\left(0,f_{r}\left(h,t\right)+\gamma -f_{r}\left(h^{\prime },t^{\prime }\right)\right)\right)\right] \end{array}$$
(23)$$\begin{array}{c} L=\lambda L_{p}+\left(1-\lambda \right)L_{k} \end{array}$$

For both tasks $L_{p}$ and $L_{k}$, the overall loss function of the RCKFM model is shown in Equation (Equation 23), and the weights of the loss values of the two learning tasks are adjusted by the parameter λ.

### 3.4. Aggregation of KL Divergence Ratings

After processing with the FM model, we can predict the ratings of each user for all movies. If the features of the trained entity have already been adjusted through a backdoor, then the features are debiased and users can get more recommendations. However, we have to consider that not all users have variable interests: some users have very singular interests and are often inclined towards one or two types of movies. In this case, we prefer to use features that were trained before backdoor adjustment to predict ratings, so we use the debiased before-and-after features to train the FM model to predict score results and merge the two score results. We introduce the KL divergence specifically and judge whether the user’s interests are varied according to the time dimension. If interests are varied, the final merged score tends towards the score after debiasing; conversely, if interests are concentrated, the score tends towards the score before debiasing. This ensures that the final recommendations are more in line with the personalized needs of the users.

In a recommendation system, a user’s interest may drift over time, resulting in a difference in interest towards projects at a specific time node compared to before. In such cases, we need to measure whether the user’s interest distribution among project types is prone to changes, using some indicator. This assists us in determining if recommendations made for the user are influenced by bias.

We choose the symmetric KL scattering [21] with the user–item interaction time as the axis, and the flow of using KL scattering for the RCKFM model is shown in Figure 11:

Specifically, we are dividing the historical interaction sequence into two symmetric parts based on the timestamps of user–item interactions. For each part, we use an equation to calculate the historical distribution of the project categories, then we use the distance between these two distributions to calculate the KL divergence, thus obtaining Formula (24),
(24)$$\begin{array}{cc} \eta_{u}= & KL\left(d_{u}^{1}|d_{u}^{2}\right)+KL\left(d_{u}^{2}|d_{u}^{1}\right) \end{array}$$
(25)$$\begin{array}{cc} = & \sum_{n=1}^{N}p_{u}^{1}\left(g_{n}\right)log\frac{p_{u}^{1}\left(g_{n}\right)}{p_{u}^{2}\left(g_{n}\right)}+\sum_{n=1}^{N}p_{u}^{2}\left(g_{n}\right)log\frac{p_{u}^{2}\left(g_{n}\right)}{p_{u}^{1}\left(g_{n}\right)} \end{array}$$
where $\eta_{u}$ denotes the final KL scatter value for that user, $d_{u}^{1}=\left[p_{u}^{1}\left(g_{1}\right),\ldots ,p_{u}^{1}\left(g_{N}\right)\right],$$d_{u}^{2}=\left[p_{u}^{2}\left(g_{1}\right),\ldots ,p_{u}^{2}\left(g_{N}\right)\right]$. Larger values of symmetric KL scatter indicate that users are more likely to change their interests and receive recommendations, i.e., backdoor adjustments are needed, and vice versa, no backdoor adjustments are needed.

For the final $\eta_{u}$, we utilize the traditional FM model and the backdoor-adjustment-based FM model to predict the final score, and we utilize the final scatter for the adaptive fusion of scores. Specifically, it is necessary to train the recommender model to compute P(Y|U = *u*, I = *i*) and P(Y|do(U = *u*), I = *i*), then automatically fuse their scores section, the effect of the backdoor adjustment, to obtain Equation (Equation 26) as follows:(26)$$\begin{array}{c} Y_{u,i}=\left(1-\hat{\eta_{u}}\right)\times Y_{u,i}^{FM}+\hat{\eta_{u}}\times Y_{u,i}^{RCKFM} \end{array}$$
where $Y_{u,i}$ is the predicted score of user *u* with item *i*, $Y_{u,i}^{FM}$ and $Y_{u,i}^{RCKFM}$ are the predicted scores of the regular recommendation model and the joint training model of RCKFM, and $\hat{\eta_{u}}$ is computed as follows in Equation (Equation 27):(27)$$\begin{array}{c} \hat{\eta_{u}}={\left(\frac{\eta_{u}-\eta_{min}}{\eta_{max}-\eta_{min}}\right)}^{a} \end{array}$$
where the normalized $\hat{\eta_{u}}\in \left[0,1\right]$, $\eta_{min}$ and $\eta_{max}$ are the minimum and maximum symmetric KL dispersion for all users, respectively, and a is a hyperparameter that can be used to control the tendency of the model’s prediction by human intervention, where the smaller a is, then the more the model’s prediction will tend to be the one that is backdoor-adjusted by causal inference.

## 4. Experiments and Results

For this section, we conducted quantitative experiments on the top-10 recommendations. In this experiment, we used the publicly available MovieLens-1M dataset and the Douban dataset from Kaggle for training. At the same time, we compared our model with multiple baseline models.

### 4.1. Data Set

As we designed the RCKFM model as an enhanced version of the CoFM model, a more reasonable and effective comparison between the two was pivotal. Therefore, we referred to the CoFM model and used the widely acknowledged public film dataset MovieLens-1M. Additionally, to broaden our comparison range, we also utilized another popular public film dataset—the Douban dataset.

MovieLens-1M comprises three files—user, movie, and rating. Notably, the user information encompasses aspects like the user ID, gender, age, profession, and zip code. The movie information includes attributes such as the movie ID, movie name, and genre while the rating information consists of the user ID, movie ID, rating, and timestamp. Its simplistic data structure and comprehensive content make it extremely valuable for research purposes in the realm of recommendation algorithms.

Similarly, the Douban dataset includes three files—user, movie, and rating. The user information in this dataset has extended attribute dimensions, such as user ID, residential area, time of joining, personality signature, and UID. The movie information includes the movie ID, movie name, director, content, actor, country, release time, language, original address, rating number, rating, genre tag, full name with the release year, category ID, and MID. The rating information includes the UID, MID, rating, timestamp, tag, category ID, and rating number ID. Both in terms of data volume and the dimensions of the data, the Douban dataset surpasses the MovieLens-1M, thereby enhancing its utilization value significantly.

To utilize both datasets they needed a level of preprocessing. Firstly, we filtered and cleaned the low-frequency items from the user–item interaction dataset, and we removed users who had given an exceedingly low number of ratings, which leads to poor representation (i.e., uncommon entities in the corresponding knowledge graph). Simultaneously, to enhance the data quality, we applied a 50-kernal filtration to the MovieLens-1M dataset—a larger user base with fewer movies too, and a 20-kernal filtration to the Douban dataset, which has fewer users but a larger collection of movies. It is important to note that the user attributes and movie attributes of the Douban dataset contain many redundant fields. These were filtered, eliminated, and replaced during data cleaning.

Secondly, we processed the historical interaction information in the dataset into a sequence of triples. Through the embedding process, we constructed negative samples from the existing positive samples by randomly replacing the items in the tuples. Then, it became necessary to match the mapping relationship between user items and entities in the knowledge graph, whereby, post-matching, the fused vectors were ingested into the collaborative filtering model—vital for joint model training.

Finally, the historical user–item interaction information in the dataset was processed on the time axis into a format that aided in calculating the KL divergence. This process simplified subsequent calculations of scatter and the fusion of predictive scores.

Table 1 shows the basic statistical information in the processed dataset. Since the sparsity of the user interaction items reached 94.64% and 98.55%, processing with the factorizer FM model to solve the problem of data sparsity could achieve very good results.

### 4.2. Baseline

We chose some fusion models of graph embedding and collaborative filtering as baseline models to compare with our proposed RCKFM model. It was necessary to use the factorization machine (FM) [14] model as the baseline method for our model, as the FM model is applied in the RCKFM model’s recommendation module. In 2018, Zhang et al. [23] proposed an integrated method of graph embedding recommendation model that is a typical fusion method. This method uses a heterogeneous graph as an external knowledge base for recommendations, thus improving the performance of the recommendation system. We chose the CoFM model based on the graph embedding recommendation algorithm as one of the baselines. This model integrates the TransE graph embedding model and the factorization machine (FM) model. For this paper, we also designed and implemented the RFM model to compare the advantages of the RCKFM model. The RFM model is a fusion model of the TransR graph embedding model and the factorization machine (FM) model. The RCKFM model, compared to the RFM model, adds causal reasoning backdoor adjustment and KL divergence personalized recommendation, so we could reflect the advantages of our designed RCKFM model by comparing the two. Meanwhile, we compared it to the current best model, FairCo [24], which introduces an error term to control the fairness of risk exposure between project groups, with a ranked list sorted by relevance to compute the error term.

### 4.3. Evaluation Metrics

In order to demonstrate the performance of the recommendation algorithms, for this chapter we used several relevant recommendation metrics as the main basis for evaluating the results. For this chapter we used all the items in the test set for recommendation, and these items were used as possible recommendation options to the user. The factor decomposer recommended the top N items as the final top recommendation to the user (the N value can be customized; the work in this paper focused on the top-10 recommendations with N of 10).

Precision@N: We defined precision as the ratio of the number of user-preferred items in the top-10 recommendations to the total number of recommendations (N). For the experimental results of multiple users in the test set, we calculated the average precision of all users as the final definition of precision. The range of its value was 0–1, and, usually, the higher the value, the better the performance. Precision@N is often abbreviated to P@N, and the calculation formula is as shown in Formula (28):
(28)$$\begin{array}{c} Precision@N=\frac{TP@N}{TP@N+FP@N} \end{array}$$
where the meanings of TP and FP are as shown in Table 2. The following formula is equivalent to this.Recall@N: We defined recall as the ratio of the number of user-preferred items in the top-10 recommendations to the total number of user-preferred items (that is, the percentage of successful recommendations of user-preferred items). Similarly, we calculated the average recall of all users as the final definition of recall. The range of its value was 0–1, and, usually, the higher the value, the better the performance. Recall@N is often abbreviated to R@N, and the calculation formula is as shown in Formula (29):
(29)$$\begin{array}{c} Recall@N=\frac{TP@N}{TP@N+FN@N} \end{array}$$NDCG@N: We defined normalized discounted cumulative gain as the cumulative benefit calculated for the first N positions, which had been normalized considering the position information. As before, we calculated the average normalized discounted cumulative gain of all users as the final definition for it. The range of its value was 0–1, and, usually, the higher the value, the better the performance. NDCG@N is often abbreviated to N@N, and the calculation formula is as shown in Formula (30):
(30)$$\begin{array}{c} NDCG@N=\frac{1}{M}\sum_{i=1}^{M}\frac{1}{log_{2}\left(p_{i}+1\right)} \end{array}$$
where M is the total number of users, p∈[1,N] denotes the computation of the first N positions, $p_{i}$ is the position of the i-th user’s real access value in the recommendation list, and $p_{i}$ tends to infinity if the value does not exist in the recommendation list.Hit Rate@N: We defined hit rate as the ratio of the number of users successfully recommended to the total number of users (if the recommendation list contained a user’s preferred item, then this user was defined as being successfully recommended). We also calculated the average hit rate of all users as the final definition of hit rate. The value range was 0–1, and, usually, the larger the value, the better the performance. Hit Rate@N is often abbreviated as HR@N, and its calculation formula is as shown in Formula (31):
(31)$$\begin{array}{c} HR@N=\frac{USER_{rate}}{USER_{test}} \end{array}$$
where $USER_{rate}$ denotes the number of users whose real recommended items in the test set appeared in the result of the recommendation list, and $USER_{test}$ denotes the total number of users in the test set. That is to say, in the overall model finally obtained, in the recommendation list top N, as long as the item is matched with the recommendation of the test set, the user will be regarded as the successful user of the recommendation.

## 5. Result

In order to explore the performance of the RCKFM model, its performance metrics with respect to each baseline model were calculated. Based on the experimental results shown in Table 3 (the best performance indicators of each model after repeated experiments), the following conclusions can be drawn:Compared with the benchmark model we adopted, the RCKFM model performs better on the dataset, and the evaluation indicators surpass the benchmark model, showing stronger competitiveness.In terms of performance improvement, compared with the results of the basic model CoFM, the improvement in various indicators of RCKFM on both datasets ranges from 3.17% to 6.81%.To compare the TransR method based on graph embedding, we compared the CoFM model and the RFM model. Their essential difference lies in the Trans series graph embedding method. After using the TransR model to embed relationships and entities into different spaces, we found that the indices of the model improved.RCKFM is the main model in this paper. By comparing with the RFM model, the RCKFM model works better than the RFM model, indicating that the RCKFM model incorporating TransR has a significant improvement in the performance of the model by incorporating the computation of causal inference backdoor adjustment and KL scatter.

In order to optimize the performance of the model, in terms of experimental parameters, using RCKFM’s top-10 model as a benchmark, for this chapter we set up several sets of controlled variable experiments, using the controlled variable method, based on which the most optimal parameters were obtained.

The optimal parameters of the experiments in initial training were: entity relationship embedding, 64 dimensions, and batch size, 512 times.

Firstly, for this chapter we conducted experiments on the dimension hyperparameters of the embedding vectors trained in the model. In order to balance the relationship between model performance and training time, the embedding dimension was set below 128 dimensions. For this chapter, the dimension vector sizes were set to 16, 32, 64, and 128, and four sets of repeated experiments were set up on these two datasets. The results of the repeated experiments were averaged to obtain the experimental results shown in Figure 12. In this set of experiments, the batch size was fixed to 512.

By analyzing the experimental results in Figure 12, it can be concluded that when the embedding dimension is 16 dimensions, the model does not fit the required features well, the overall performance index of the model is the lowest, and the model training is not good; when the vector dimension is 64, the model’s final top-10 recommendation index effect reaches the best. This indicates that in the RCKFM model with less than 128-dimensional embedding, the top-10 recommendation list obtained from the 64-dimensional embedded feature vectors trained by the factor factorizer FM is the best recommendation effect. Moreover, the low variation of the model’s hit rate indicates that the overall performance of the model is better and the recommendation hit effect is more stable.

At the same time, on the basis of the experiments with fixed batch size and different dimensions, the feature vectors obtained from different embedding dimensions were put into FM training, the effects of the feature vectors obtained from different embedding dimensions of the TransR graph embedding model on the training accuracy of the FM model were analyzed, and the experimental results as shown in Figure 13 were obtained.

By analyzing the experimental results in Figure 13, it can be ascertained that when the embedding dimension is 16 dimensions, the model does not capture the features of the entity particularly well, and the model training accuracy is the lowest; when the embedding dimension is 64 dimensions, the embedded feature vector results obtained from the TransR model can be put into the FM model to obtain a higher training accuracy; the overall fluctuation of the training accuracy is lower than 0.05, which indicates that the performance of the model is relatively stable, but 64-dimensional TransR embedding is recommended for the best model results.

Secondly, for this chapter we investigated the effect of different batch sizes on the model performance. According to the above experimental experience, in order to make the model have better performance on item recommendation, we needed to fix the dimensions of the TransR graph embedding training as 64 dimensions, take 128 as the minimum batch size, set different batch sizes as 128, 256, 512, and 1024, respectively, and set four groups of repetitive experiments on the dataset to take the average of the repetitive experimental results, so as to obtain the experimental results shown in Figure 14:

By analyzing the experimental results in Figure 14, we could ascertain that the overall performance of the model got better and better with the increase in batch size: when the batch size was 512, the top-10 recommendation indicators obtained after model training were basically at the highest value; when the batch size was 1024, the model only hit the same rate, as the batch size is 512 on the Douban dataset. Meanwhile, comparing all the indicators, we could ascertain that the difference between the maximum and minimum value of the indicator axis was very small, which indicates that the batch size had less influence on the recommendation indicators and that the indicator change was more stable. At the same time, comparing all the indicators, we could ascertain a small difference between the maximum and minimum values of the indicator axes, which indicates that the batch size had less influence on the recommendation indicators and that the indicator change amplitude was more stable, which also indicates that the batch size was not a factor that subjectively affected the recommendation effect.

Meanwhile, on the basis of the above experiments, for this chapter we fixed the TransR graph embedding to the embedding dimension of 64 dimensions and the number of embedding training rounds to one hundred rounds. The average time consumption statistics of each round were carried out according to the different batch sizes of the embedded graphs, and the experimental results shown in Figure 15 were obtained after repeated experiments of many rounds:

By analyzing the experimental results in Figure 15, we could ascertain that the TransR model was embedded when the batch size was 128, that if the batch size decreased then the total number of batches would increase, that the result obtained after repeated experiments was the maximum training time consumption, that when the batch size was 1024 the total number of batches was small, and that the corresponding result was the lowest training time consumption. In summary, the larger the training batch size, the shorter the time required for model training. Of course, it is not a case of ’the larger the batch size the better’: if the batch is too large, this may also affect the model performance. Therefore, when weighing the training time and training metrics, the model works best with a batch size of 512.

In summary, the best training situation for the RCKFM joint training model is to determine the embedding dimension to be 64 dimensions at the time of TransR embedding and the training batch size to be 512. In this case, not only the training time can be greatly reduced, but also the performance of the model can be greatly improved, which is a win–win training parameter obtained from the repeated experiments with controlled variables conducted in this chapter. Overall, the performance of the RCKFM model is mainly related to the embedding dimension, the overall performance is relatively stable, the impact of different batch sizes on the experimental performance indicators is relatively weak, and the RCKFM model is highly competitive in terms of recommendation performance.

## 6. Conclusions

Based on the CoFM model, this paper implements optimization improvements, enhances model performance, and distinguishes it from the RFM model, a model improved from CoFM. A new recommendation model acting upon TransR, backdoor adjustment of causal inference, KL divergence, and fusion factor decomposition, termed RCKFM, is proposed. The RCKFM model replaces the graph embedding-based TranE model with the TransR model, capable of addressing 1-N, N-1, and N-N problems, based on the CoFM model. Concurrently, it uses the backdoor adjustment of causal inference to reconstruct user features, eliminate the prejudice suffered by user-feature values from the users’ project history interaction, and integrates the traditional FM recommendation and the final predicted score of RCKFM through symmetric KL divergence to realize an adaptive personalized user–project recommendation. The research carried out experiments on the public Movieslens-1M dataset and the Douban dataset on Kaggle’s official website using the FM, CoFM, RFM, FairCo, Diversity, and RCKFM models. The better performance of the RCKFM model shows that the TransR model can better identify the rich knowledge provided by the knowledge graph and effectively extract feature relationships from the data. It also indicates that after integrating causal inference backdoor adjustments and KL divergence, the model is closer to personalized recommendations, providing better effects. In terms of energy efficiency, our model utilizes bi-directional graph embedding to obtain low-dimensional features, which avoids the time loss associated with secondary user coding and reduces the time overhead of collaborative filtering training.

In the future, our focus will be on utilizing neural network structural models, such as multi-layer perceptron (MLP) and graph convolutional networks (GCN), to effectively fuse entities and neighborhoods. Additionally, we aim to integrate domain-specific and item-specific information, such as movie entities, along with associated directors, actors, countries, etc., to improve feature representation. Drawing from our research on large language models (LLMs), our future endeavors will involve exploring the fusion of LLMs with knowledge graphs and extracting entity features from them, with the goal of achieving enhanced recommendations. Meanwhile, we plan to design timed experiments to compare the energy efficiency of the model proposed in this paper with other models to demonstrate the energy efficiency of the recommender system.

Our study aims to enhance the effectiveness of personalized collaborative filtering recommendation systems, targeting primarily academic peers, industry practitioners, and decision makers. To disseminate our research findings to these audiences, we will actively participate in relevant academic conferences and share our research discoveries through academic forums and social media platforms. We believe that through these efforts, our research outcomes will positively impact both academia and industry, fostering progress and advancement in the field of recommendation systems. In case our initial dissemination measures fail to reach the intended target audiences, we have prepared a contingency plan. This plan includes exploring additional dissemination channels, such as industry-specific events and professional networks, as well as collaborating with key stakeholders to ensure wider reach and uptake of our research findings.

## Figures and Tables

**Figure 1 entropy-26-00371-f001:**
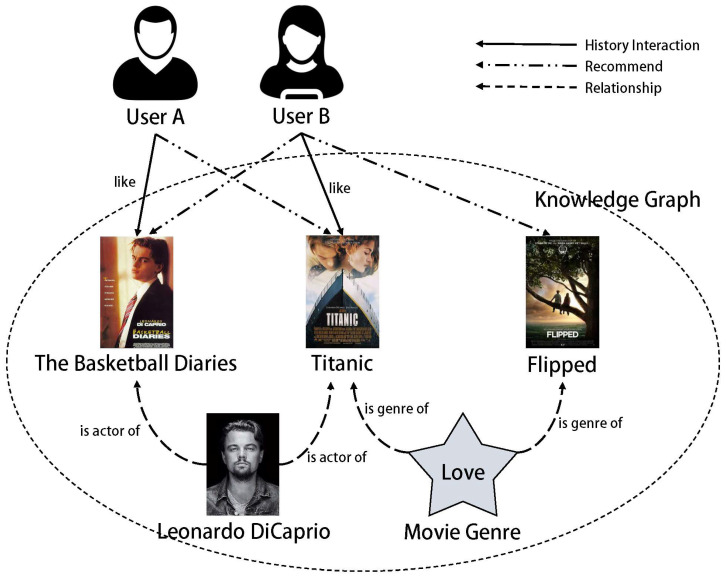
User interaction with the knowledge graph.

**Figure 2 entropy-26-00371-f002:**
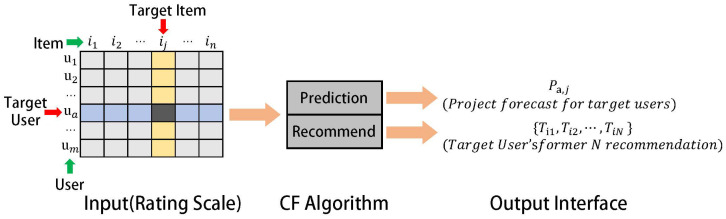
Basic idea of collaborative filtering using similarity calculation.

**Figure 3 entropy-26-00371-f003:**
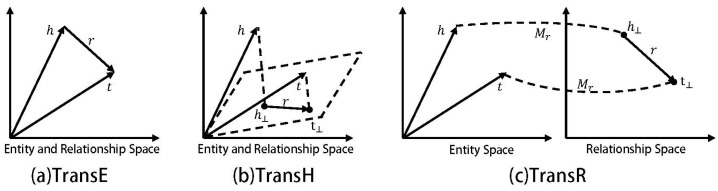
Three commonly used distance-based translation models.

**Figure 4 entropy-26-00371-f004:**
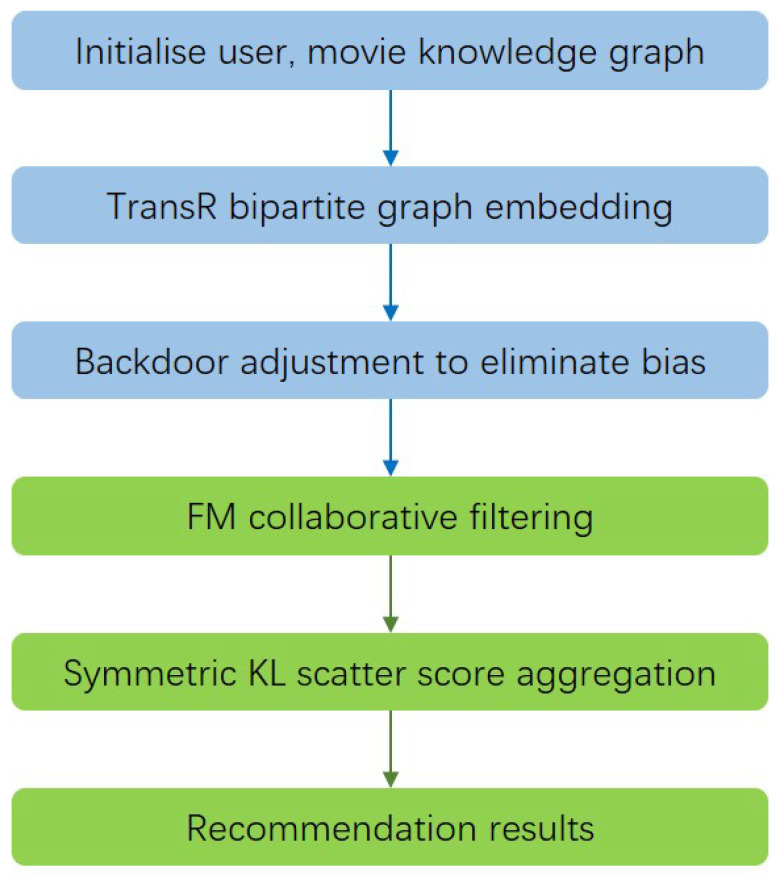
RCKFM modeling flows.

**Figure 5 entropy-26-00371-f005:**
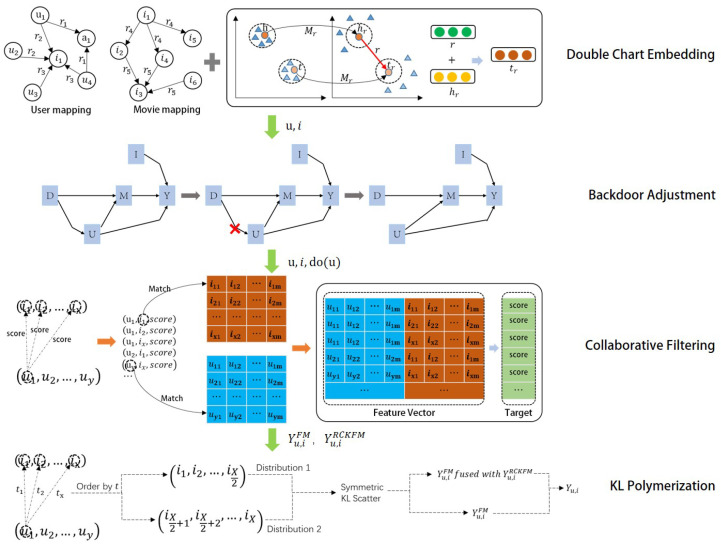
Framework of the RCKFM model.

**Figure 6 entropy-26-00371-f006:**
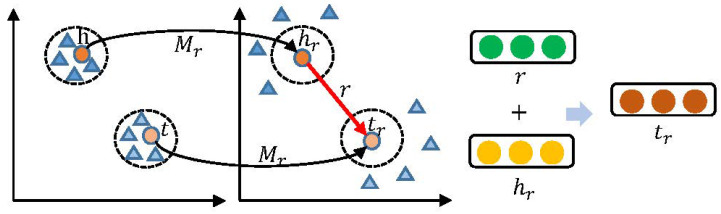
TransR model.

**Figure 7 entropy-26-00371-f007:**
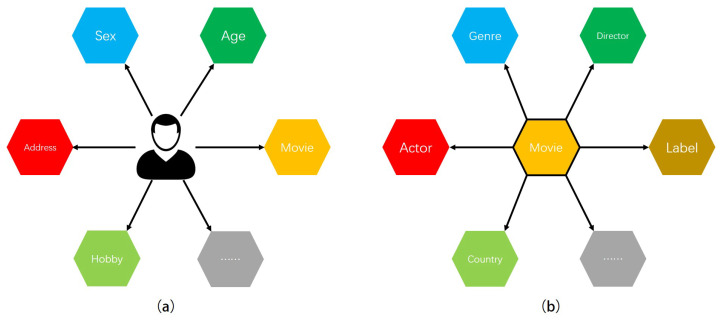
Bipartite graph construction.

**Figure 8 entropy-26-00371-f008:**
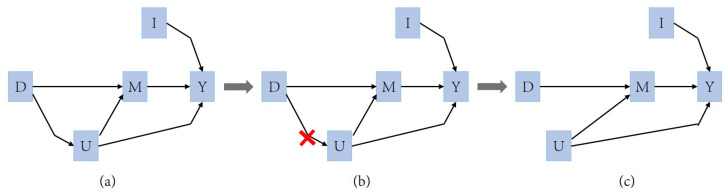
Backdoor adjustment of causal inference.

**Figure 9 entropy-26-00371-f009:**
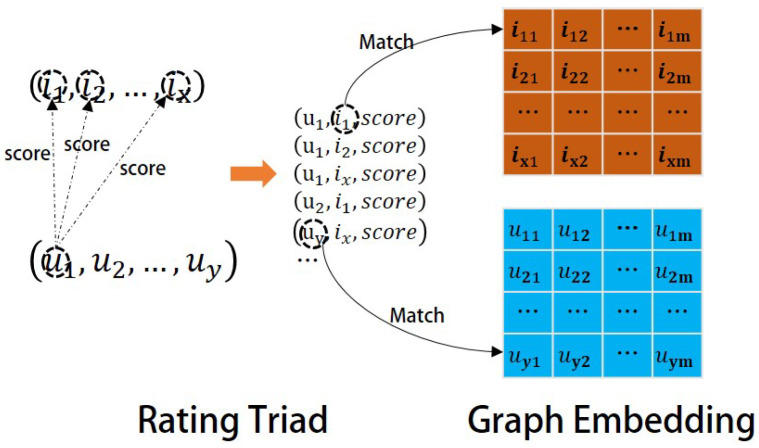
Characteristic matching of users and projects.

**Figure 10 entropy-26-00371-f010:**
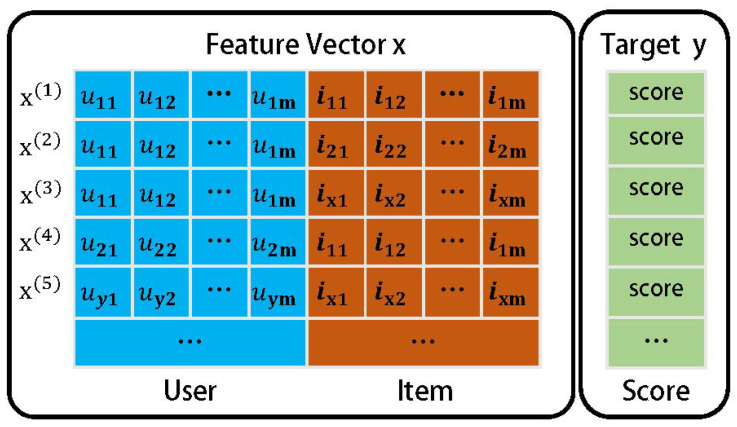
Factor decomposer FM model.

**Figure 11 entropy-26-00371-f011:**
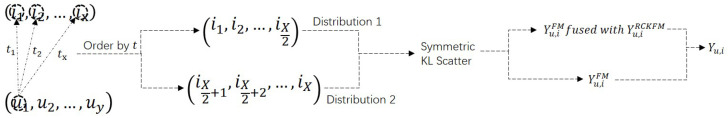
TRFM framework (flowchart of KL dispersion).

**Figure 12 entropy-26-00371-f012:**
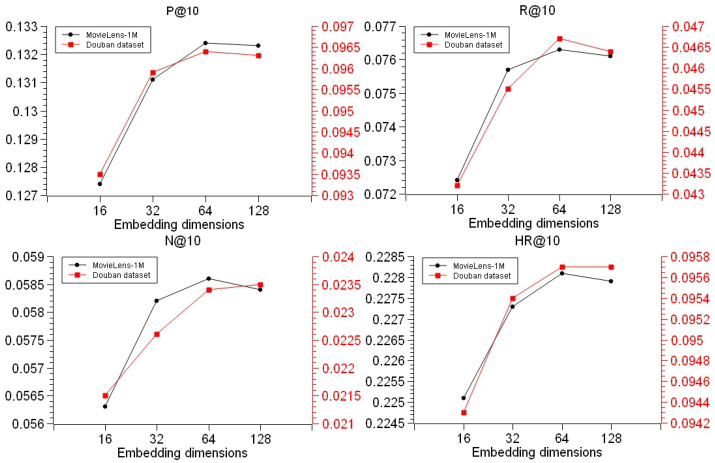
Effect of different embedding vector dimensions on model performance.

**Figure 13 entropy-26-00371-f013:**
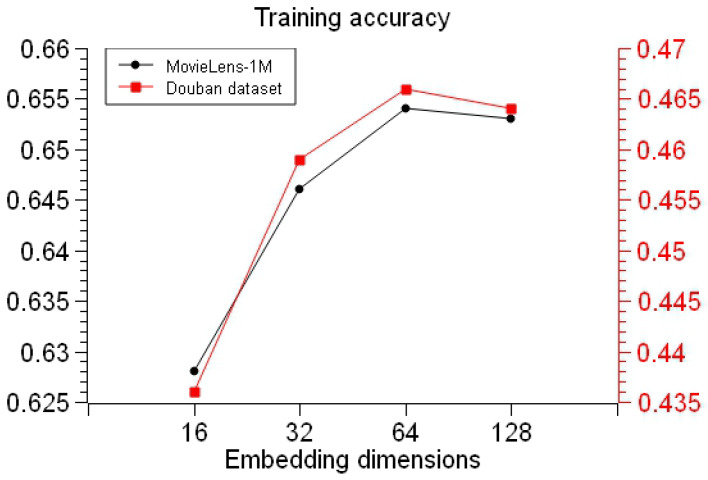
Effect of different embedding dimensions of TransR on FM training accuracy.

**Figure 14 entropy-26-00371-f014:**
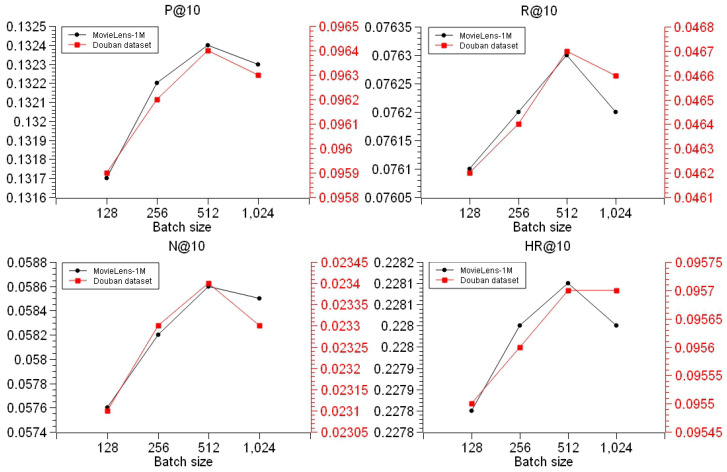
Effect of different batch sizes on model performance.

**Figure 15 entropy-26-00371-f015:**
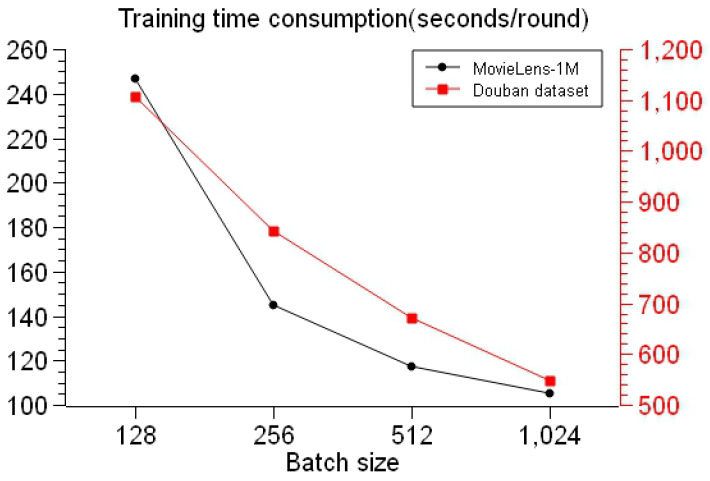
Time consumption for different batch sizes.

**Table 1 entropy-26-00371-t001:** Basic data information.

		MovieLens-1M	Douban Dataset
User–Item Interaction	User	4297	2519
	Item	3883	34,893
	Relationship	6	17
	Rating	893,578	1,276,928
	Average of Ratings	210	507
	Sparsity	94.64%	98.55%

**Table 2 entropy-26-00371-t002:** Experimental indicators.

	System Recommendations	System Not Recommended
Like	True Positive Rate (TP)	False Negative Rate (FN)
Dislike	False Positive Rate (FP)	True Negative Rate (TN)

**Table 3 entropy-26-00371-t003:** Top-10 experiment results.

	MovieLens-1M				Douban Dataset			
**Method**	**P@10**	**R@10**	**N@10**	**HR@10**	**P@10**	**R@10**	**N@10**	**HR@10**
FM	0.1241	0.0682	0.0549	0.2084	0.0914	0.0418	0.0213	0.0853
CoFM	0.1272	0.0728	0.0568	0.2165	0.0925	0.0443	0.0221	0.0896
RFM	0.1297	0.0745	0.0576	0.2215	0.0955	0.0452	0.0225	0.0929
FairCo	0.1306	0.0754	0.0581	0.2243	0.0959	0.0457	0.0228	0.0935
RCKFM	0.1324	0.0763	0.0586	0.2281	0.0964	0.0467	0.0234	0.0957
Improve	4.09%	4.81%	3.17%	5.36%	4.22%	5.42%	5.88%	6.81%

## Data Availability

The data presented in this study are available on request from the corresponding author.

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
