# Peer review of "A Personalized Collaborative Filtering Recommendation System Based on Bi-Graph Embedding and Causal Reasoning"

_entropy, 2024, doi:10.3390/e26050371_

Round 1
Reviewer 1 Report
Comments and Suggestions for Authors
Dear Authors,
thanks for submitting your work to Entropy: The paper is relevant and of interest for many readers. I would like to give some comments in order to improve the paper:
General Comments:
1.) The description is a bit lengthy in almost all parts, it can be more compact and the text - in particular the description of the overall model - can be more formal.
2.) If possible please give a comparison to the state-of-the-art in the literature. As you mentioned, the used databases seem to be very common and therefore results in the literature are available.
Minor comments:
- Figure captions are very brief , e.g. Fig 2 "The basic idea of CF" .. I suggest to extend and improve the figure captions
- eq (1) : use a function form of description for standard deviation , as for covariance, so \sigma(x)
- SEc 2.2 HerÄ™ "Borderline Diary" is mentioned - but it can not be found in Figure 1
- equation (29) is nice but I think the "equivalent" equation, king - man = queen - women gives a better interpretation. BTW: it would be good to learn more about the used methods and quality of the used embeddings
- In the results section always, single results , perhaps averaged are given. It would be good to see results of multiple trials and to see how stable these results are, see for instance Figure 13, right panels, here the curves are oscillating (upper panel for the movie lens lower panel for the Douban data base) Are there any reasons for that?
Reviewer 2 Report
Comments and Suggestions for Authors
In this manuscript the authors discuss about a personalized collaborative filtering recommendation system that is based on bi-graph embedding and causal reasoning. They firstly introduce the related domain, providing the challenges and the difficulties of current integrated recommendation algorithms. Afterwards, they are introducing their research, concluding to the experiments they have performed.
Overall a nice manuscript that should consider the following:
1. Provide in the abstract the reason behind this research with clear examples
2. Avoid providing in the introduction the numbers of the references without introducing their content
3. Provide in the introduction a list of limitations and research gaps that led you to this research
4. Provide a paragraph indicating the structure of the rest of the document at the end of the introduction
5. With regards to the related work, provide a chapter with regards to the energy efficiency of such recommendation systems along with their usability and outcomes. You should also refer to the following studies:
a. Karabetian, Andreas, et al. "An Environmentally-sustainable Dimensioning Workbench towards Dynamic Resource Allocation in Cloud-computing Environments." 2022 13th International Conference on Information, Intelligence, Systems & Applications (IISA). IEEE, 2022.
b. Chicaiza, Janneth, and Priscila Valdiviezo-Diaz. "A comprehensive survey of knowledge graph-based recommender systems: Technologies, development, and contributions." Information 12.6 (2021): 232.
c. Lin, Rongping, et al. "Energy-efficient computation offloading in collaborative edge computing." IEEE Internet of Things Journal 9.21 (2022): 21305-21322.
d. Gu, Xiaohui, and Guoan Zhang. "Energy-efficient computation offloading for vehicular edge computing networks." Computer Communications 166 (2021): 244-253.
6. Provide a figure in section 3 indicating the flow of the steps that someone should follow to replicate this work
7. For the used dataset provide a snapshot indicating its content and its variables – are there any limitations for it?
8. Have you performed any data preprocessing techniques to the used datasets prior to their usage?
9. Provide the limitations and the lessons learned for this work – have you also made any assumption for it? If so, provide them as well
10. Who are the receivers of this work? How are you going to disseminate your results to them?
11. Provide your next goals as well as a timeplan towards achieving them at the conclusion section
Round 2
Reviewer 1 Report
Comments and Suggestions for Authors
Thanks for revising the paper!
Author Response
Dear Editor and Reviewer 1:
Thank you very much for taking the time to review our manuscript. We sincerely appreciate your thorough evaluation and valuable feedback.
Best regards.
Xiaoli Huang
Reviewer 2 Report
Comments and Suggestions for Authors
The authors have addressed some of the comments, but the below still remain an open issue:
(1) There are still missing clear examples in the abstract proving the reason behind this research
(2) Provide in the introduction a list of limitations and research gaps that led you to this research, along with your research assumptions since it is still not well covered
(3) I insist that the authors to provide a chapter with regards to the energy efficiency of such recommendation systems along with their usability and outcomes. The below research studies should be studied in deeper detail and properly referred to, since the authors have done a similar work:
a. Karabetian, Andreas, et al. "An Environmentally-sustainable Dimensioning Workbench towards Dynamic Resource Allocation in Cloud-computing Environments." 2022 13th International Conference on Information, Intelligence, Systems & Applications (IISA). IEEE, 2022.
b. Lin, Rongping, et al. "Energy-efficient computation offloading in collaborative edge computing." IEEE Internet of Things Journal 9.21 (2022): 21305-21322.
c. Gu, Xiaohui, and Guoan Zhang. "Energy-efficient computation offloading for vehicular edge computing networks." Computer Communications 166 (2021): 244- 253.
(4) With regards to the related preprocessing, does it also consider any data transformation or data harmonization technique? This is not clear yet .
(5) It is still unclear the way that the research outcomes will be disseminated and communicate to the external audience.
Round 3
Reviewer 2 Report
Comments and Suggestions for Authors
The authors addressed almost most of the comments, but the following should be for sure addressed:
(1) I insist that the authors to provide a chapter with regards to the energy efficiency of such recommendation systems along with their usability and outcomes. If this is not possible, provide it in a paragraph as a future step, that energy efficiency will be also considered, since environmental sustainability is among the top priorities in the ICT world. The below research studies should be studied in deeper detail and properly referred to, since the authors have done a similar work:
a. Karabetian, Andreas, et al. "An Environmentally-sustainable Dimensioning Workbench towards Dynamic Resource Allocation in Cloud-computing Environments." 2022 13th International Conference on Information, Intelligence, Systems & Applications (IISA). IEEE, 2022.
b.Lin, Rongping, et al. "Energy-efficient computation offloading in collaborative edge computing." IEEE Internet of Things Journal 9.21 (2022): 21305-21322.
c. Gu, Xiaohui, and Guoan Zhang. "Energy-efficient computation offloading for vehicular edge computing networks." Computer Communications 166 (2021): 244- 253.
(2) What could happen if the dissemination measures would not reach the targeted audience? Is there any backup plan for this?
